# NOISE TOLERANCE OF DISTRIBUTIONALLY ROBUST LEARNING

**Ramzi Dakhmouche**

Institute of Mathematics, EPFL, Switzerland

Computational Engineering Lab, Empa, Switzerland

`ramzi.dakhmouche@epfl.ch`

**Ivan Lunati**

Computational Engineering Lab, Empa, Switzerland

`ivan.lunati@empa.ch`

**Hossein Gorji**

Computational Engineering Lab, Empa, Switzerland

`mohammadhossein.gorji@empa.ch`

## ABSTRACT

Given the importance of building robust machine learning models, considerable efforts have recently been put into developing training strategies that achieve robustness to outliers and adversarial attacks. Yet, a major aspect that remains an open problem is systematic robustness to global forms of noise such as those that come from measurements and quantization. Hence, we propose in this work an approach to train regression models from data with additive forms of noise, leveraging the Wasserstein distance as a loss function. Importantly, our approach is agnostic to the model structure, unlike the increasingly popular Wasserstein Distributionally Robust Learning paradigm (`WDRL`) which, we show, does not achieve improved robustness when the regression function is not convex or Lipschitz. We provide a theoretical analysis of the scaling of the regression functions in terms of the variance of the noise, for both formulations and show consistency of the proposed loss function. Lastly, we conclude with numerical experiments on physical PDE benchmarks and electric grid data, demonstrating competitive performance with an order of magnitude reduction in computational cost.

## 1 INTRODUCTION

In real-world applications, collected data is often tainted with different forms of noise. Whether it is sensor noise in engineering systems or measurement uncertainty in biological experiments, such noise usually demands costly and time-consuming pre-processing steps, before meaningful results can be extracted using predictive machine learning algorithms. In order to streamline that process, different robust learning approaches have been proposed with a focus on robustness to outliers and adversarial attacks (Mohajerin Esfahani and Kuhn, 2018; Steinhardt et al., 2018; Bai et al., 2023; Levine and Feizi, 2020). Most of such strategies rely on augmenting the data with adversarial examples (Goodfellow et al., 2014; Madry et al., 2018) or designing suitable loss regularization techniques (Dong et al., 2020). However, for more global forms of noise, which are commonly encountered in practice, these approaches face both statistical and practical limitations. In the case of data augmentation, the limitations are inherent to its design, while adversarial regularization often targets bounded perturbations, thereby overlooking standard noise models that arise in real-world settings. In contrast, the increasingly popular paradigm of Wasserstein Distributionally Robust Learning (`WDRL`) (Mohajerin Esfahani and Kuhn, 2018; Shafieezadeh-Abadeh et al., 2019; Gao et al., 2024) represents a more general framework that allows for arbitrary perturbations, and is more theoretically appealing while leading to competitive performance. Yet, there seems to be a gap in the literature when it comes to robustness properties of `WDRL` with respect to global forms of noise, as pointed out by Hu et al. (2020) for instance. In this work, we address this question in a regression setting from multiple perspectives:

1. We study the global robustness properties of the popular `WDRL` formulation, through a theoretical and numerical analysis of its scaling in terms of the variance of the noise.

2. Notably, we show that `WDRL` may fail to improve the performance when the regression functions are neither Lipschitz nor convex.

3. To address this limitation, we propose a simple yet powerful robust learning approach that is agnostic to the structure of regression functions, enabling more expressive models. We further provide a theoretical analysis of its dependence on the variance of the noise.

4. We numerically demonstrate the performance of our proposed approach through various physical problem benchmarks and electric grid usage time series data.

The rest of the paper is organized as follows. In Section 2, we review the regression setting. We explore the shortcomings of `WDRL`, which has not been investigated before, in Section 3. Section 4 introduces the novel regression approach, where we establish its main theoretical properties in Section 5. We justify the improved scaling of the introduced regression method with respect to the empirical risk minimization and `WDRL` in Proposition 5.1. Section 6 is devoted to the numerical results, where the competitive performance of our method is demonstrated. Finally, Section 7 discusses the limitations, outlook and concluding remarks.

## 1.1 RELATED WORKS

**Denoising and Filtering.** Extensive research has been conducted on denoising and filtering techniques, ranging from Kalman filtering (Ito and Xiong, 2000) and wavelet denoising (Sardy et al., 2001) to deep learning based methods (Jain and Seung, 2008; Xing and Egiazarian, 2021; Krull et al., 2019; Lehtinen et al., 2018). For a comprehensive overview in the context of image data modalities, see Elad et al. (2023). However, most of these approaches require low noise data, focus on Gaussian noise distributions or require an explicit noise model. Additionally, they introduce costly pre-processing steps that must be performed prior to the modeling. In contrast, we propose an approach that directly trains competitive models from noisy data, eliminating the need for extensive pre-processing.

**Adversarial Defense.** Early works introduced techniques to augment the training data with adversarial examples (Goodfellow et al., 2014; Madry et al., 2018), leveraging the expressive power of neural networks to improve robustness. Building on this, several regularization techniques such as entropic regularization (Dong et al., 2020) and adversarial weight perturbation (Wu et al., 2020) have been proposed, further enhancing their performance. In parallel, *certified robustness* approaches have focused on quantifying the proportion of samples that remain robust to arbitrary perturbations within a given bound (Tjeng et al., 2019; Raghunathan et al., 2018; Dathathri et al., 2020). However, these techniques often lead to overly conservative models, which can degrade performance in the presence of global noise perturbations (Bai et al., 2023).

**Distributionally Robust Optimization.** It is concerned with minimizing the worst-case loss over a given set of distributions (Mohajerin Esfahani and Kuhn, 2018; Föllmer and Weber, 2015; Blanchet and Murthy, 2019), which is formally expressed as the minimax problem

$$\inf_{\theta \in \Theta} \sup_{Q \in \mathcal{P}} \mathbb{E}_Q \left[ \ell_\theta(Z) \right]$$

where the supremum is taken over a suitably chosen class of distributions $Q \in \mathcal{P}$. Recent focus (Shafieezadeh Abadeh et al., 2015; Staib and Jegelka, 2017; Shafieezadeh-Abadeh et al., 2019; Chen and Paschalidis, 2018; Bartl et al., 2021; Gao et al., 2024; Phan et al., 2023) has been given to the formulation with Wasserstein ambiguity set $\mathcal{P} = B_\delta(P)$, which is the ball centered at the empirical distribution $P$ with radius $\delta$ under the Wasserstein distance, leading to `WDRL`. See (Gao and Kleywegt, 2023) for a discussion on theoretical advantages of this choice. `WDRL` has demonstrated remarkable performance in out-of-sample linear regression (Mohajerin Esfahani and Kuhn, 2018) and classification (Shafieezadeh-Abadeh et al., 2019) tasks, as well as in defending against adversarial attacks (Bai et al., 2023; Bui et al., 2022) on neural networks. For classification problems, another highly effective, although computationally costly, approach was proposed by Zhai et al. (2021). In contrast, we consider robustness to unbounded forms of noise encountered in regression, which to the best of our knowledge, has not been much investigated for deep learning models (Liu et al., 2024).

## 2   PROBLEM SETTING & BACKGROUND

We denote by $\mathcal{P}_2(\Omega)$ the set of probability measures over $\Omega$ with finite variance. The Euclidean 2-norm is given by $\| \cdot \|_2$. Consider the realizations $(X_i)_{i \leq n}$ from a measure in $\mathcal{P}_2(\Omega)$, we employ $m[(X_i)_{i \leq n}] = 1/n \sum_{i=1}^{n} \delta_{X_i}$, as the shorthand for empirical measure. Consider the regression task of predicting response variables $y \in \mathcal{Y}$ from input features $x \in \mathcal{X}$. Given a class of regression functions $\{ f_\theta, \theta \in \mathbb{R}^d \}$ and data samples $\{ X_i, Y_i \}_{i \leq n}$ from an underlying distribution $\mu_{(X,Y)} \in \mathcal{P}_2(\mathcal{X} \times \mathcal{Y})$ with $f_\theta : \mathcal{X} \to \mathcal{Y}$, the standard goal is to find a model $\theta \in \mathbb{R}^d$ that minimizes the empirical risk

$$\hat{\theta}_{\text{MSE}} \in \arg\min_{\theta \in \mathbb{R}^d} \frac{1}{n} \sum_{i=1}^{n} \| Y_i - f_\theta(X_i) \|_2^2.$$

In Wasserstein Distributionally Robust Learning (Ben-Tal et al., 2013; Bartl et al., 2021), the aim instead is to minimize a stronger form of the empirical risk

$$\hat{\theta}_{\text{WDRL}} \in \arg\min_{\theta \in \mathbb{R}^d} \sup_{\substack{(X,Y) \sim \mu \\ \mathcal{W}_2(\mu, \hat{\mu}) \leq \delta}} \mathbb{E}_\mu[\ell(Y, f_\theta(X))] , \tag{1}$$

where $\hat{\mu} \in \mathcal{P}_2(\mathcal{X} \times \mathcal{Y})$ is the empirical distribution over the training data, $\ell$ a pre-chosen loss function, and $\mathcal{W}_2$ the 2-Wasserstein distance given by

$$\mathcal{W}_2(\mu, \hat{\mu}) = \min_{\pi \in \Pi(\mu, \hat{\mu})} \int_{(\mathcal{X} \times \mathcal{Y})^2} \| \alpha - \beta \|_2^2 \, \mathrm{d}\pi(\alpha, \beta)$$

with $\Pi(\mu, \hat{\mu})$ the set of couplings between the probability measures $\mu$ and $\hat{\mu}$, and $\alpha, \beta \in \mathcal{P}_2(\mathcal{X} \times \mathcal{Y})$.

In this work, we focus on the setup where the response samples $(Y_i + \sigma \varepsilon_i)_{i \leq n}$ are tainted with independent and identically distributed noise with variance $\sigma^2$, with the objective of training deep learning models that are the least sensitive to the noise level $\sigma$.

## 3   DRAWBACKS OF WDRL

In order to compute the objective function of WDRL, certain structural assumptions on the regression functions $f_\theta, \theta \in \mathbb{R}^d$ as well as the loss function $\ell$, must be imposed. This is necessary since the original formulation (1) involves solving an infinite dimensional optimization problem, which is generally intractable. For that matter, two main settings have been proposed:

(a) Assuming that the function $\ell_\theta : (x, y) \mapsto \ell(y, f_\theta(x))$ is a finite maximum of concave functions;

(b) Assuming that the function $\ell_\theta : (x, y) \mapsto \ell(y, f_\theta(x))$ is Lipschitz continuous.

In either cases, the objective function given in (1) can be rewritten (Mohajerin Esfahani and Kuhn, 2018; Shafieezadeh Abadeh et al., 2015) under the tractable form[1]

$$d_2((Y_i)_{i \leq n}, (f_\theta(X_i))_{i \leq n}) := \sup_{\substack{(X,Y) \sim \mu \\ \mathcal{W}_2(\mu, \hat{\mu}) \leq \delta}} \mathbb{E}_\mu[\ell(Y, f_\theta(X))]$$

$$= \inf_{\lambda \geq 0} \left[ \lambda \delta + \frac{1}{n} \sum_{i=1}^{n} \sup_{(\xi_1, \xi_2) \in \mathcal{X} \times \mathcal{Y}} \left\{ \ell(\xi_1 - f_\theta(\xi_2)) - \lambda \| Y_i - \xi_1 \|_2^2 - \lambda \| X_i - \xi_2 \|_2^2 \right\} \right],$$

where the optimal solutions $\lambda^\star(\theta)$ and $(\xi_1^\star(\theta), \xi_2^\star(\theta))$ are reached for all $\theta \in \mathbb{R}^d$. Yet, to satisfy $(a)$ or $(b)$ in a regression setting where the data distributions have unbounded domains, one typically needs to set $\ell(y, x) = |y - x|$ and to enforce structural properties of convexity or Lipschitzness on the neural network models, therefore, reducing their expressive power. A natural question that emerges is whether using the tractable expression of $d_2$ as a loss function, regardless of whether the equality holds, can improve the robustness of the neural network models. We provide a negative answer to this

---

[1](Blanchet and Murthy, 2019) propose a more general condition for the equality to hold, but leave the question of existence of optimizers, which is essential here, open.

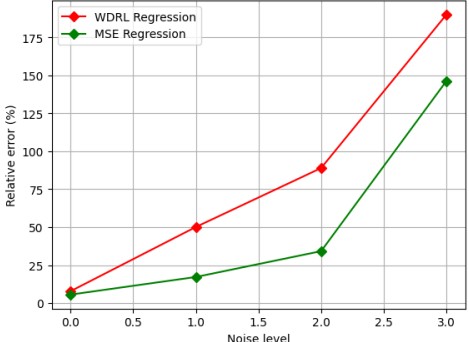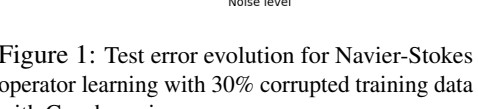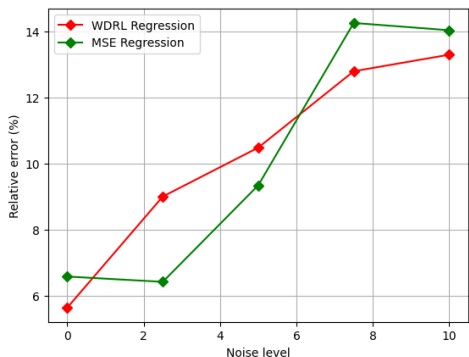

Figure 1: Test error evolution for Navier-Stokes operator learning with 30% corrupted training data with Cauchy noise

Figure 2: Test error evolution for Navier-Stokes operator learning with 30% corrupted training data with Gaussian noise

question by exploring the behavior of $d_2$ in training a convolutional neural operator (CNO) (Raonic et al., 2024) to solve the two-dimensional Navier-Stokes equation. In particular, we estimate the operator that maps the initial condition ($T = 0$), represented as an image, to the final state ($T = 1$). To this end, we train the `WDRL` regression model employing a stochastic descent- ascent algorithm, exploring the model behavior as the noise level increases. We use the hyperparameters optimized by the authors who proposed the CNO architecture (Raonic et al., 2023). We obtain the results shown in Figs. 1 and 2, for both Gaussian and heavy-tail noise distributions, respectively. For the latter case, we use the standard Cauchy distribution, where $\sigma$ represents the scale parameter, as a Cauchy random variable does not have finite variance due to the heavy tails. We examine model performance via the mean absolute relative error (MAE). Our results indicate that under heavy-tail noise, `WDRL` training performs significantly worse than the standard `MSE` training. In the Gaussian noise setting, both lead to comparable results, without noticeable improvement from `WDRL`. This is in contrast with the novel regression approach introduced in the follow-up section, whose performance on this setting is demonstrated in Section 6. Note that this limitation of `WDRL` has not been raised so far, to the best of our knowledge, mainly because previous works focused on image classification where the data domains are bounded, honouring the Lipschitz property.

## 4 WASSERSTEIN BATCH MATCHING

The key idea behind our approach is to relax the strict matching between features $X_i$ and their responses $Y_i + \sigma\varepsilon_i$ for $i \in I_p$, where $I_p$ denotes the index set of a training batch. The motivation for this relaxation is that, in the presence of noise, the observed response $Y_i + \sigma\varepsilon_i$ already deviates from the true response $Y_i$. Consequently, if the batch elements are close enough, allowing features to match responses without a fixed correspondence yields more robust estimates while reducing the sensitivity of the loss function to the noise. We provide in the following a consistency result as well as an asymptotic analysis of the `WBM` loss function.

### 4.1 FORMULATION & CONSISTENCY

The formal way to implement this idea is to compute the Wasserstein distance between the empirical distributions of the predictions $(f_\theta(X_i))_{i \in I_p}$ and the responses $(Y_i)_{i \in I_p}$, leading to Wasserstein Batch Matching (`WBM`) regression

$$\hat{\theta}_{\text{WBM}} \in \arg\min_{\theta \in \mathbb{R}^d} \sum_{p \geq 1} \mathcal{W}_2(m[(Y_i)_{i \in I_p}], m[(f_\theta(X_i))_{i \in I_p}]) \,,$$

instead of the Mean Squared Error (`MSE`) regression. We illustrate the `WBM` idea in figure 3. For our setting of empirical distributions, note that the Wasserstein distance reduces to

$$\mathcal{W}_2(m[(Y_i)_{i \in I_p}], m[(f_\theta(X_i))_{i \in I_p}]) = \min_{P \in C} \left\langle P, M_{((Y_i)_{i \in I_p}, (f_\theta(X_i))_{i \in I_p})} \right\rangle \,,$$

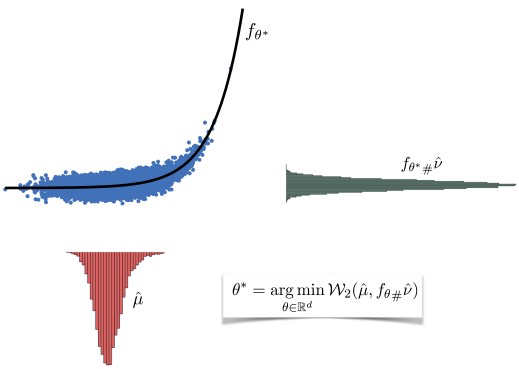

Figure 3: Wasserstein Batch Matching illustration. The regression through cloud of points in a batch (depicted by blue) is tackled by finding optimal map, depicted by black line, between distributions of $(X_i)_i$ and $(Y_i + \sigma\epsilon)_i$, shown by red and green histograms.

where $M_{((Y_i)_{i \in I_p},(f_\theta(X_i))_{i \in I_p})} = \left( \|Y_i - f_\theta(X_j)\|_2^2 \right)_{i,j \in I_p}$ is the matrix of the pairwise norms between the predictions and the target values, and $C$ the set of coupling matrices of dimension $\#I_p$. As a sanity check, we show in proposition (1) below that, asymptotically such a matching scheme recovers any continuously differentiable bandlimited function, among its co-monotonic functions from its samples, in the noise-free regime. The proof is reported in Appendix (A).

**Proposition 4.1.** *(Consistency)*
*Let $f : \mathbb{R}^d \longrightarrow \mathbb{R}$ be a continuously differentiable and integrable function with compactly supported Fourier transform and let $(f(x_{\phi(i)}))_{i \leq n}$ be its values sampled at ordered points $(x_i)_{i \leq n}$, where $\phi$ is an unknown permutation preserving the batch partition. Then, given a fixed batch size and an arbitrary amount of samples, $f$ is completely characterized by minimizing*

$$\min_{g \in \mathcal{G}} \sum_{p \geq 1} \mathcal{W}_2(m[(f(x_i))_{i \in I_p}], m[(g(x_j))_{j \in I_p}]) \,,$$

*where $\{I_p, p \geq 1\}$ is the finite collection of batch index sets and $\mathcal{G}$ the set of continuously differentiable and integrable bandlimited functions that are co-monotonic with $f$.*

**Remark.** (**Complexity**) From a computational complexity perspective, training with WBM involves solving a linear program at each training step, which costs $O(s)$, where $s = dim(\mathcal{Y})$ is the dimension of the response space $\mathcal{Y}$. However, this is independent of the strucutre of the regression functions. On the other hand, WDRL involves solving a minimax problem which is in $O(s^3)$ when the function $\ell_\theta$ is convex-concave. However, in the absence of this structure, the problem can become arbitrarily hard.

**Remark.** (**Differentiability**) The proposed WBM loss is differentiable with respect to the regression parameters $\theta \in \mathbb{R}^d$ by the envelope theorem (Bonnans and Shapiro, 2013), which makes it suitable for training deep learning models.

## 5 NOISE SCALING ANALYSIS

To investigate the theoretical advantage of the WBM regression, we examine the robustness of the learned function with respect to the noise variance. First, we provide a sensitivity analysis of the scaling of the WDRL- and WBM- loss functions in terms of the variance of the noise, and contrast their scaling with that of the MSE loss. Next, we investigate and discuss the consequences of this scaling on the learned regression functions.

### 5.1 NOISE SCALING OF LOSS FUNCTIONS

An important feature of the introduced WBM loss is its scaling with respect to the noise which is derived in the following. The proof is reported in Appendix (B).

**Proposition 5.1.** *(**Noise Scaling**) Assume*[2] *the response variables* $(Y_i)_{i \leq n}$ *to be normalized and let* $\sigma^2$ *denote the variance of the output noise. Then, for* $\sigma \in (0, 1)$, *we have for the* WBM *loss*

$$\frac{1}{2}\mathcal{W}_2\left(m[(Y_i + \sigma\varepsilon_i)_{i \in I_p}], m[(f_\theta(X_i))_{i \in I_p}]\right) - \frac{1}{2}\mathcal{W}_2\left(m[(Y_i)_{i \in I_p}], m[(f_\theta(X_i))_{i \in I_p}]\right) =$$
$$\sum_{i,j \in I_p} \left[(Y_i - f_\theta(X_j)) - (Y_i - f_\theta(X_j))^3\right] P_{i,j}\sigma\varepsilon_i + O(\sigma^2),$$

*where* $(P_{i,j})_{i,j}$ *is a constant coupling matrix. For the* WDRL *loss, we have*

$$|d_2\left((Y_i + \sigma\varepsilon_i)_{i \in I_p}, (f_\theta(X_i))_{i \in I_p}\right) - d_2\left(Y_i)_{i \in I_p}, (f_\theta(X_i))_{i \in I_p}\right)| \leq \sigma \cdot \frac{1}{n}\sum_{i=1}^{n}|\varepsilon_i| + o(\sigma).$$

*where we assumed the regression function* $f_\theta$ *to be Lipschitz and continuously differentiable.*

**Remark.** To put the previous bounds into perspective, consider the noise scaling of the MSE loss

$$d_{\text{MSE}}\left((Y_i + \sigma\varepsilon_i)_{i \in I_p}, (f_\theta(X_i))_{i \in I_p}\right) - d_{\text{MSE}}\left((Y_i)_{i \in I_p}, (f_\theta(X_i))_{i \in I_p}\right) = \frac{2\sigma}{\#I_p}\sum_{i \in I_p}(Y_i - f_\theta(X_i))\,\varepsilon_i$$
$$+ O(\sigma^2),$$

where we took $\mathcal{Y} = \mathbb{R}$, for simplicity of presentation and $d_{\text{MSE}}$ is the MSE loss, given by

$$d_{\text{MSE}}((Y_i)_{i \in I_p}, (X_i)_{i \in I_p}) = \frac{1}{\#I_p}\sum_{i \in I_p}\|Y_i - X_i\|_2^2.$$

Comparing the scaling of WBM loss with the one resulting from MSE, we observe that for small deviations between $(Y_i)_{i \in I_p}$ and $(f_\theta(X_i))_{i \in I_p}$, the prefactor of the linear term is smaller in the former. This reduction arises because the WBM loss penalizes the prefactor through a cubic term when the deviation is less than unity. This is not surprising, since WBM performs regression over the infimum of all possible couplings between the regression function and the response, leading to a diminished dependency of the WBM loss compared to MSE. This theoretical insight is further confirmed by numerical experiments demonstrated in Section 6. In the following, we clarify how the derived scalings can have implications on the learned function $f_\theta$ for $\theta \in \mathbb{R}^d$.

## 5.2 Noise Scaling of Regression Functions

To estimate the effect of the scaling of the loss on the learned parameters defining the estimated regression functions, we consider the invariant probability measures of the Markov chains defined by the Stochastic Gradient Descent (SGD) iterates, given in the MSE case by

$$\theta_{k+1} = \theta_k - \eta\nabla_\theta\left(\frac{1}{\#I_p}\sum_{i \in I_p}(Y_i - f_\theta(X_i))^2\right) - \nabla_\theta\left(\frac{\sigma}{\#I_p}\sum_{i \in I_p}(Y_i - f_\theta(X_i))\,\varepsilon_i\right).$$

This is motivated by the fact that SGD is a standard way to solve the regression problem, when training deep learning models. Furthermore, this perspective has the advantage of taking into account other sources of randomness, encountered in model training. Yet, to keep the presentation concise, following constant step-size stochastic approximation literature (Dieuleveut et al., 2020; Park et al., 2022; Vlatakis-Gkaragkounis et al., 2024), we assume in the following the loss functions to be strongly convex. This assumption can be relaxed when considering decreasing step-sizes algorithms (Atchadé et al., 2017; Ben Arous et al., 2022; Arous et al., 2021). In this instance, this assumption translates to considering regularized versions of the loss functions with reproducing Hilbert kernel space hypotheses classes (Wainwright, 2019). The following results address regularized MSE and WBM, and are based on applying results by (Dieuleveut et al., 2020). We postpone their proofs to Appendix (D) and (E) respectively.

---

[2] without loss of generality

**Corollary 5.2.** *Assume the* MSE *and* WBM *loss functions (denoted generically as $\ell_\theta$) to be strongly convex. Then, under suitable regularity assumptions (detailed in Appendix D), the averaged optimization iterates*

$$\bar{\theta}_k = \frac{1}{k+1} \sum_{j=0}^{k} \theta_k$$

*converge to a unique stationary distribution $\pi_\eta$ with expectation $\bar{\theta}_\eta = \int_{\mathbb{R}^d} \alpha \, d\pi_\eta(\alpha)$ such that*

$$\bar{\theta}_\eta - \theta^\star = \eta \left(\nabla^2 \ell_{\theta^\star}\right)^{-1} \left(\nabla^3 \ell_{\theta^\star}\right) A(\theta^\star) V(\theta^\star) + O(\eta^2)$$

*where $\theta^\star$ is the optimum of the noise free respective problem, $A = \left(\nabla^2 \ell_{\theta^\star} \otimes I + I \otimes \nabla^2 \ell_{\theta^\star}\right)^{-1}$ and $V(\theta^\star) = \mathbb{E}\left[(\nabla_\sigma \ell_{\theta^\star})^{\otimes 2}\right]$.*

**Remark.** We can see the effect of the first order coefficient of the loss functions in terms of the noise level in $V(\theta^\star)$. For the WDRL loss, the gradient iterates have a non-centered bias term making convergence not achievable in general. We derive bounds on the resulting learnt parameters in corollary E.1 (see Appendix E).

## 6    NUMERICAL RESULTS

We demonstrate the performance of WBM regression on two important practical problems: operator learning for PDEs and electric grid usage forecasting. Evaluation on test data is carried out using the mean absolute error (MAE) in all displayed figures, where we compare models trained with the MSE to those trained with WBM. We focus on this setting, since the state-of-the-art method for practical noise regimes WDRL was shown in section 3 to underperform ERM. For completeness though, we report comparison of WBM to divergence-based methods CVaR DRO and Chi-Sq DRO (Duchi and Namkoong, 2021), in figure 7. As for GCDRO (Liu et al., 2024), it is based on kNN graph construction, which is known to perform poorly on high-dimensional data (Radovanović et al., 2009), which we focus on in this work. We explore the robustness properties of WBM both to standard Gaussian and heavy-tail Cauchy noise. Heavy-tail noise is present in many real-world applications such as vibration sensors for intelligent monitoring, power consumption sensors, and LIDAR systems. It comes from transient events, sudden extreme changes such as short circuits, or atmospheric noise which exhibits heavy-tails. Additionally, we explore robustness to distribution shift properties, by training on (almost) noise-free data and testing on noisy data. Such a use-case is encountered in practice, when a model is developed based on cleaned data before being deployed on real data. The results are averages over 13 samples. Code available at code.

### 6.1    LEARNING OF PDES

PDEs model a wide range of physical and engineering problems and feature a rich set of dynamical processes that illustrate the performance of machine learning models on a wide range of practical regression problems. For that matter, we demonstrate the performance of WBM on two extensively used PDEs: the wave equation and the Navier-Stokes equation. More precisely, we consider the corresponding recently proposed benchmarks in Raonic et al. (2024), where the task consists of learning operators mapping initial conditions ($T = 0$), represented as images, to the final state reached by the system, e.g., corresponding to ($T = 1$). The underlying images represent two-variable functions sampled at a given resolution. We train convolutional neural operators (Raonic et al., 2023), which have been proposed as featuring robustness properties notably to change in resolution. We set the hyperparameters optimized in (Raonic et al., 2023), and keep the same for WBM training, except the batch size for which we explore different values. The convolutional neural operator architecture is based on mapping the sampled images back to function space using the Whittaker-Shannon interpolation formula (Raonic et al., 2023). We display the results in Figs. 4, 5, 6 and 7. We note that, WBM regression consistently outperforms MSE regression. In particular, while both MSE and WDRL regressions indicate significant errors in Navier-Stokes operator learning subject to the Cauchy noise, as shown in Fig. 6, the introduced WBM regression demonstrates notable robustness. Furthermore, we note that WBM regression outperforms divergence-based DRO methods, as shown in Fig. 7, while being significantly cheaper computationally. Indeed, note that WBM does require additional hyper-parameter tuning leading to at least 10-fold computational training gain compared to CVaR and Chi-Sq DRO. We report additional numerical results and error bars in Appendix F.

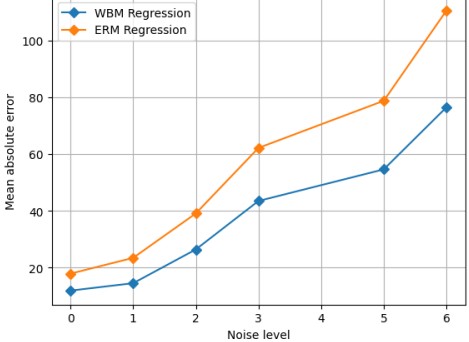

Figure 4: Test error evolution for Navier-Stokes operator learning; 30% corrupted test data with Cauchy noise

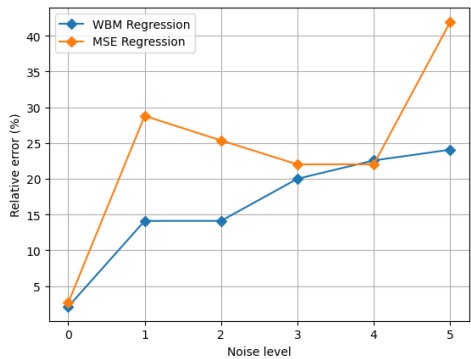

Figure 5: Test error evolution for wave equation operator learning; 30% corrupted training data with Gaussian noise

## 6.2 ELECTRIC GRID USAGE FORECASTING

Predicting electric load is an important and timely problem, especially given the increasing share of renewable energy sources in grid networks. We employ the recently proposed state-of-the-art model `TSMixer` (Chen et al.), to forecast electric transformer usage from the popular ETDataset (Zhou et al., 2021; Ilbert et al., 2024). `TSMixer` is based on mixing operations via stacking multi-layer perceptions. We train the model with input sequence length of 336 and prediction sequence length of 96. We utilize the hyperparameters proposed by the authors, except the number of training epochs which we reduce to a single swap over the data. This is justified by the fact that we compare the model against itself trained with different loss functions. Hence, the comparison point can be chosen in a flexible way. We display the results in Figs. 8 and 9, where `WBM` outperforms `MSE`, especially in the case of Cauchy noise, suggesting its performance extends across high-dimensional learning problems.

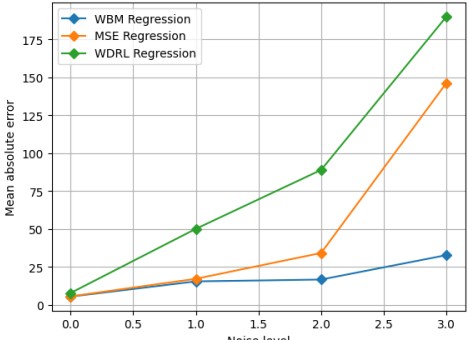

Figure 6: Test error evolution for Navier-Stokes operator learning; 30% corrupted training data with Cauchy noise

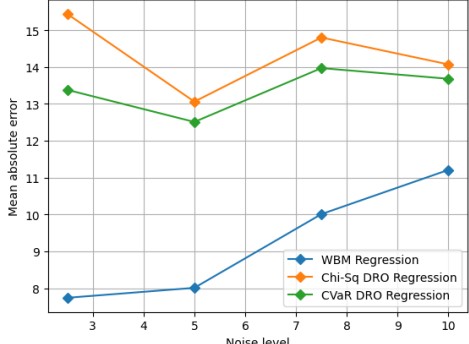

Figure 7: Test error evolution for Navier-Stokes operator learning; 30% corrupted training data with Gaussian noise

## 7 DISCUSSION

**Limitations & Outlook.** Although `WBM` demonstrates competitive performance, it has certain limitations. First, as indicated by Proposition 4.1, `WBM` requires the underlying function to be sufficiently regular. As a consequence, it struggles in operator learning tasks involving discontinuities, such as shock wave problems. Additionally, in the low-noise regime, `WBM` may exhibit slight underperformance. This can be attributed to weak matching underlying `WBM`, which, while beneficial in noisy settings, is suboptimal when perfect knowledge of the responses is available. That being

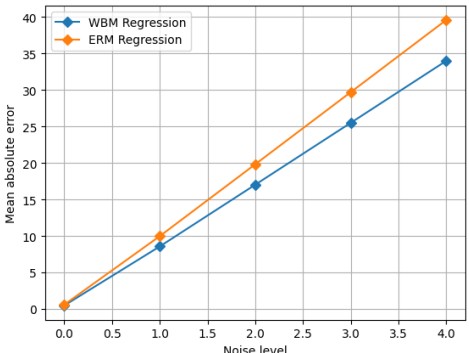

Figure 8: Test error evolution for electric time series forecasting; 30% corrupted test data with Cauchy noise

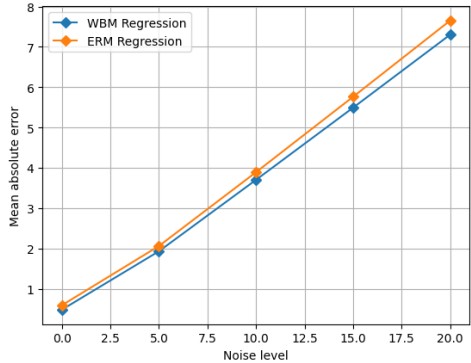

Figure 9: Test error evolution for electric time series forecasting; 30% corrupted test data with Gaussian noise

said, most real-world data is noisy and there exists a fundamental limit of learning algorithms when it comes to performing well both on clean and noisy data. Despite the limitations, `WBM` introduces a novel perspective on robust learning, paving the way for future research into robustness against unbounded, correlated, or structured forms of noise.

**Conclusion.** We considered the open problem of robustness to global forms of additive noise and proposed an efficient learning approach `WBM`, overcoming the drawbacks of current robust learning methods. We investigated the scaling of the introduced regression along with other approaches with respect to noise levels, offering a theoretical justification for the gains achieved by `WBM`. Furthermore, we demonstrated the practical performance of `WBM` via several numerical experiments involving PDE operators and electrical time series forecasting. We believe this work paves the way for robust learning methods that streamline the costly data pre-processing step, while advancing the development of reliable machine learning models.

## ACKNOWLEDGMENTS

This work was supported by the Swiss National Science Foundation under grant No. 212876. We acknowledge computational resources from the Swiss National Supercomputing Centre CSCS.

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

## A  PROOF OF PROPOSITION 4.1: CONSISTENCY

Let's first assume that $\mathcal{X} = \mathcal{Y} = \mathbb{R}$. Let $f, g \in \mathcal{G}$. Given that $g$ is continuously differentiable, it has bounded variations on every compact, that is for all $a, b \in \mathbb{R}$ such that $a < b$, we have

$$\sup_{pr \in Pr} \sum_{i=1}^{n_{pr}} |g(x_{i+1}) - g(x_i)| < +\infty \, ,$$

where the supremum is taken over the set

$$\left\{ pr = \{x_0, \dots, x_{n_{pr}}\} \mid pr \text{ is a partition of } [a, b] \text{ satisfying } x_i \leq x_{i+1} \text{ for } 0 \leq i \leq n_{pr} - 1 \} \right\}$$

This implies that there exists a partition of the feature space into intervals of lengths $(\delta_n)_{n \in \mathbb{N}}$ such that $g$ is monotonous on every interval. The same holds for $f$. Hence, we consider the partition formed by $I_i \cap J_j$ where $(I_i)_{i \in \mathbb{N}}$ and $(J_j)_{j \in \mathbb{N}}$ are the chosen partitions for $f$ and $g$ respectively. We denote by $(\delta_n)_{n \in \mathbb{N}}$ the new lengths. Since, $f$ is bandlimited, let the support of its Fourier transform be included in $[-B, B]$ with $B > 0$. We can choose $(\delta_n)_n$ such that $\delta_n \leq \frac{1}{B}$ for all $n \in \mathbb{N}$. Furthermore, we can sample each interval a number of times equal to the prefixed batch size. Since, $f$ satisfies

$$\min_{g \in \mathcal{G}} \sum_{p \geq 1} \mathcal{W}_2 \left( m[(f(x_i))_{i \in I_p}], m[(g(x_j))_{j \in I_p}] \right) = \sum_{p \geq 1} \mathcal{W}_2 (m[(f(x_i))_{i \in I_p}], m[(f(x_j))_{j \in I_p}])$$
$$= 0,$$

we know that a minimizer $g \in \mathcal{G}$ of

$$g \mapsto \sum_{p \geq 1} \mathcal{W}_2 \left( m[(f(x_i))_{i \in I_p}], m[(g(x_j))_{j \in I_p}] \right)$$

must satisfy

$$\forall p \geq 1, \quad \mathcal{W}_2(m[(f(x_i))_{i \in I_p}], m[(g(x_j))_{j \in I_p}]) = 0 \, .$$

Furthermore, since $f$ and $g$ are co-monotonic, the Wasserstein matching recovers the true matching. Last, by the Shannon sampling theorem we conclude $g$ is equal $f$. As for generalization to the multi-dimensional setting, the co-monotonicity property is expressed through $\nabla f \cdot \nabla g \geq 0$. Intervals are replaced by hyperrectangles (Cartesian products of intervals) and the matching argument carries on over hyperrectangles. Last, the bounded variation property carries on as an implication of Mean Value Inequality.

## B  PROOF OF PROPOSITION 5.1: NOISE SCALING

### B.1  WASSERSTEIN BATCH MATCHING

For simplicity of the presentation, we take $\mathcal{Y} = \mathbb{R}$. The proof relies on establishing the scaling for the Sinkhorn regularized version $W_\lambda$ of the Wasserstein distance, then concluding by taking the limit as $\lambda \to +\infty$. This is motivated by the fact that $W_\lambda$ is strictly convex with respect to the coupling matrix $P$, hence it allows for a richer characterization of the optimal coupling matrix $P^\star$ and its differentiability properties. First, recall that for $\lambda > 0$,

$$W_\lambda(a, b) = \min_{P \in U(a,b)} \langle P, M \rangle - \frac{1}{\lambda} h(P) \quad \text{with} \quad h(P) = - \sum_{i,j=1}^{n} P_{i,j}(\log P_{i,j} - 1) \, ,$$

where $M$ is the cost matrix, that depends on the data $\{X_i, Y_i + \sigma \varepsilon_i\}_{i \leq n}$, and $a$, $b$ the discrete distributions under consideration. In our case, we consider uniform $a$ and $b$. That is, $a_i = b_i = \frac{1}{\#I_p}$ for $i \leq \#I_p$. Denoting by $P_\lambda$ the solution of the above optimization problem, we have by Sinkhorn's scaling theorem (Sinkhorn and Knopp, 1967; Luise et al., 2018) the characterization

$$P_\lambda = \text{diag}(e^{\lambda \alpha^\star}) e^{-\lambda M} \text{diag}(e^{\lambda \beta^\star}) \, ,$$

where $\alpha^\star$ and $\beta^\star$ are the optimal Lagrange multipliers. We would like to compute the derivative of $P_\lambda$ with respect to $\sigma$ to deduce its sensitivity to noise for small values of $\sigma$, for e.g. $\sigma \in (0, 1)$. But, first let us establish that $\sigma \mapsto P_\lambda(\sigma)$ is differentiable. Consider the Lagrangian

$$\mathcal{L}(\sigma; \alpha, \beta) = \alpha^\top a + \beta^\top b + \sum_{i,j=1}^{n} \frac{e^{-\lambda(M_{ij} - \alpha_i - \beta_j)}}{\lambda} \, .$$

By theorem 2 in (Luise et al., 2018), it is smooth and strictly convex in $\gamma = (\alpha, \beta)$, hence for every fixed $\sigma > 0$, there exists $\gamma^\star(\sigma)$ such that $\mathcal{L}(\sigma; \gamma^\star(\sigma)) = \min_\gamma \mathcal{L}(\sigma; \gamma)$. As a result, following the proof of theorem 2 in (Luise et al., 2018), by the implicit function theorem, we deduce that $P_\lambda$ is continuously differentiable. Furthermore, denoting $\psi = \nabla_\gamma \mathcal{L}$, we have

$$\psi(\sigma; \gamma) = \begin{pmatrix} a - P_\lambda \mathbf{1} \\ b - P_\lambda^\top \mathbf{1} \end{pmatrix}$$

with

$$\psi(\sigma; \gamma^\star(\sigma)) = 0.$$

By differentiation, we get

$$\nabla_1 \psi(\sigma, \gamma^\star(\sigma)) + \nabla_\sigma \gamma^\star(\sigma) \nabla_2 \psi(\sigma, \gamma^\star(\sigma)) = 0$$

and consequently

$$\nabla_\sigma \gamma^\star(\sigma) = \begin{pmatrix} \nabla_\sigma \alpha^\star(\sigma) \\ \nabla_\sigma \beta^\star(\sigma) \end{pmatrix} = \begin{pmatrix} (2(Y_i - f_\theta(X_j)) + 2\sigma\varepsilon_i)_{i,j \leq n} \\ (2(Y_i - f_\theta(X_j)) + 2\sigma\varepsilon_i)_{i,j \leq n} \end{pmatrix}.$$

Now, we can compute

$$\frac{\mathrm{d}}{\mathrm{d}\sigma} \sum_{i,j=1}^n M_{ij}(\sigma)(P_\lambda^\star(\sigma))_{ij} = \sum_{i,j=1}^n \frac{\mathrm{d}M_{ij}(\sigma)}{\mathrm{d}\sigma}(P_\lambda^\star(\sigma))_{ij} + \sum_{i,j=1}^n M_{ij}(\sigma)\frac{\mathrm{d}(P_\lambda^\star(\sigma))_{ij}}{\mathrm{d}\sigma} =$$

$$\sum_{i,j=1}^n 2(Y_i - f_\theta(X_j))(P_\lambda^\star(\sigma))_{ij} + M(\sigma)(\nabla_\sigma \alpha^\star(\sigma)P_\lambda^\star(\sigma))_{ij} + M(\sigma)(\nabla_\sigma M_{ij}(\sigma)P_\lambda^\star(\sigma))_{ij}$$

$$+ M(\sigma)(\nabla_\sigma \beta^\star(\sigma)P_\lambda^\star(\sigma))_{ij} =$$

$$\sum_{i,j=1}^n 2(Y_i - f_\theta(X_j))(P_\lambda^\star(\sigma))_{ij} - M(\nabla_\sigma \beta^\star(\sigma)P_\lambda^\star(\sigma))_{ij} ,$$

where we discarded the terms in $O(\sigma)$. Finally, since this holds for each $\sigma \in (0, 1)$, and $P_\lambda(\sigma) \to P(\sigma)$ where $P(\sigma)$ is the optimal coupling in the non-regularized Wasserstein distance, which is continuous in $\sigma$ (see for e.g. (Bonnans and Shapiro, 2013)), we conclude by Dini's theorem that the convergence is uniform, hence by taking the limit when $\lambda \to +\infty$, of the estimate on the derivative, we get the result.

## B.2 WASSERSTIEN DISTRIBUTIONALLY ROBUST LEARNING

Under the Lipschitz assumption, we obtain the existence of optimizers for $d_2$. Such optimizers $(\lambda^\star, \xi_1^\star, \xi_2^\star)$ satisfy

$$0 \in \partial_{\lambda, \xi_1, \xi_2} d_2((Y_i + \sigma\varepsilon_i)_{i \leq n}, (f_\theta(X_i))_{i \leq n}) ,$$

where $\partial_{\lambda, \xi_1, \xi_2} d_2((Y_i + \sigma\varepsilon_i)_{i \leq n}, (f_\theta(X_i))_{i \leq n})$ is the Clarke subdifferential of $d_2$ leveraging the differentiability properties of $\bar{f}_\theta$ and the implicit function theorem, we use this inclusion to obtain the derivatives of the optimizers with respect to $\sigma$. Note that the only non-smooth part comes from taking $\ell(x, y) = |x - y|$. Specifically, we get

$$\begin{cases} 0 \in 1 - 2\lambda(\xi_1 - Y_i + \sigma\varepsilon_i) \, \partial|\cdot| \\ .0 \in [\nabla f_\theta - 2\lambda(\xi_2 - X_i)] \, \partial|\cdot| \end{cases}$$

by differentiation we get

$$\begin{cases} 0 \in -2\lambda(\frac{d\xi_1}{d\sigma} + \varepsilon_i) \, \partial|\cdot| \\ 0 \in \frac{d\xi_2}{d\sigma} \nabla^2 f_\theta(\xi_2) - 2\lambda\frac{d\xi_2}{d\sigma}\partial|\cdot| \end{cases}$$

since $\partial|\cdot| = [-1, 1]$, and leveraging the Lagrangian to differentiate $\lambda$ with respect to $\sigma$ we get

$$0 \in (\delta - \|\xi_1 - Y_i\|^2 - \|\xi_2 - X_i\|)\frac{d\lambda}{d\sigma} + \frac{1}{2\lambda}[-1, 1] .$$

Now, as we have the derivatives of the optimizers, we can differentiate

$$d_2((Y_i + \sigma\varepsilon_i)_{i \leq n}, (f_\theta(X_i))_{i \leq n}) = \lambda^\star \delta + \frac{1}{n} \sum_{i=1}^n \left\{ \ell(\xi_1^\star - f_\theta(\xi_2^\star)) - \lambda\|Y_i - \xi_1^\star\|_2^2 - \lambda\|X_i - \xi_2^\star\|_2^2 \right\}$$

leading to the result.

## C  DISCUSSION ON ASYMPTOTIC SCALING

Further insights into the relationship between WBM and MSE regression can be obtained by considering the asymptotic limit of regression problem in a batch of infinitely many samples. Specifically, we analyze the case where the feature-response pair $(X, Y) \in \mathcal{X}^2$ are random variables, satisfying the following conditions.

1. The laws of $X$ and $Y$, $\mu_{X,Y}$, are Gaussian measures with mean values $m_{X,Y}$ and covariances $\Sigma_{X,Y}$.

2. The covariance between $X$ and $Y$ is optimal, i.e., $\text{Cov}(X, Y) = \left( \Sigma_X^{1/2} \Sigma_Y \Sigma_X^{1/2} \right)^{1/2}$.

3. The regression is performed over linear functions of $\theta \in \mathbb{R}$.

In this setting, it is straight-forward to see that the two loss functions become identical. In particular, following Gelbrich bound (Gelbrich, 1990), we have

$$\mathcal{W}_2(\mu_Y, f_{\theta \#} \mu_X) \quad = \quad \|m_X - \theta m_Y\|_2^2 + \text{tr}\, \Sigma_Y + \theta^2 \text{tr}\, \Sigma_X - 2\theta \text{tr}\, \left( \Sigma_X^{1/2} \Sigma_Y \Sigma_X^{1/2} \right)^{1/2},$$

where $f_{\theta \#} \mu_X$ is push-forward of $\mu_X$ by $f_\theta$, and therefore the MSE loss becomes

$$
\begin{aligned}
\mathbb{E}\|Y - \theta X\|_2^2 \quad &= \quad \|m_X - \theta m_Y\|_2^2 + \text{tr}\, \Sigma_Y \\
&+ \quad \theta^2 \text{tr}\, \Sigma_X - 2\theta \text{tr}\, \text{Cov}(X, Y) \\
&= \quad \mathcal{W}_2(\mu_Y, f_{\theta \#} \mu_X) \,.
\end{aligned}
$$

# D PROOF OF COROLLARY 5.2

First, let us detail the assumptions. In addition to strong convexity, we assume that the loss function is five times continuously differentiable with second to fifth bounded derivatives. Moreover, we assume for the covariance matrix $C(\theta) = \mathbb{E}[(\nabla_\sigma)^{\otimes 2}]$, there exists $s, k \geq 0$ such that for all $\theta \in \mathbb{R}^d$, we have

$$\max_{i \in \{1,2,3\}} \left\| C^{(i)}(\theta) \right\| \leq M(1 + \|\theta - \theta^\star\|^k) .$$

Recall that for the MSE loss given by

$$d_{\text{MSE}} \left( (Y_i + \sigma \varepsilon_i)_{i \in I_p}, (f_\theta(X_i))_{i \in I_p} \right) = \frac{1}{\#I_p} \sum_{i \in I_p} (Y_i + \sigma \varepsilon_i - f_\theta(X_i))^2 ,$$

we have

$$\nabla_\theta d_{\text{MSE}} \left( (Y_i + \sigma \varepsilon_i)_{i \in I_p}, (f_\theta(X_i))_{i \in I_p} \right) = \nabla_\theta \left( \frac{1}{\#I_p} \sum_{i \in I_p} (Y_i - f_\theta(X_i))^2 \right)$$

$$+ \nabla_\theta \left( \frac{\sigma}{\#I_p} \sum_{i \in I_p} (Y_i - f_\theta(X_i)) \varepsilon_i \right)$$

along with

$$\nabla_\theta \left( \frac{\sigma}{\#I_p} \sum_{i \in I_p} (Y_i - f_\theta(X_i)) \varepsilon_i \right) = \frac{\sigma}{\#I_p} \sum_{i \in I_p} \varepsilon_i \nabla_\theta (Y_i - f_\theta(X_i)) ,$$

and thus

$$\mathbb{E} \left[ \nabla_\theta \left( \frac{\sigma}{\#I_p} \sum_{i \in I_p} (Y_i - f_\theta(X_i)) \varepsilon_i \right) \right] = \frac{\sigma}{\#I_p} \sum_{i \in I_p} \mathbb{E}[\varepsilon_i] \nabla_\theta (Y_i - f_\theta(X_i)) = 0 .$$

Similarly for the WBM loss, we get

$$\mathbb{E} \left[ \nabla_\theta \mathcal{W}_2 \left( m[(Y_i + \sigma \varepsilon_i)_{i \in I_p}], m[(f_\theta(X_i))_{i \in I_p}] \right) \right] = \nabla_\theta \mathcal{W}_2 \left( m[(Y_i)_{i \in I_p}], m[(f_\theta(X_i))_{i \in I_p}] \right)$$

$$+ \xi((Y_i)_{i \in I_p}, (f_\theta(X_i))_i, (\varepsilon_i)_i) ,$$

with $\mathbb{E}[\xi((Y_i)_{i \in I_p}, (f_\theta(X_i))_i, (\varepsilon_i)_i)] = 0$. Hence, by Proposition 2 and Theorem 4 in (Dieuleveut et al., 2020), we obtain the result for both loss functions.

# E PARAMETER ERROR BOUNDS

**Corollary E.1.** *Assume the loss functions to be strongly convex with constant $\kappa > 0$. Then, for $\sigma \in (0, 1)$, the WBM noise-free estimate $\theta^\star_{\text{WBM}}$ satisfies*

$$\|\theta^\star_\sigma - \theta^\star_{\text{WBM}}\| \leq \sqrt{\frac{D^1_{\text{WBM}} \sigma}{\kappa}} + \sqrt{\frac{D^2_{\text{WBM}} \sigma^2}{\kappa}}$$

*where $\theta^\star_\sigma$ is the noisy estimate and $D^1_{\text{WBM}}, D^2_{\text{WBM}} > 0$ a constant. In contrast, the noise-free MSE estimate satisfies*

$$\|\theta^\star_\sigma - \theta^\star_{\text{MSE}}\| \leq \sqrt{\frac{D^1_{\text{MSE}} \sigma}{\kappa}} + \sqrt{\frac{D^2_{\text{MSE}} \sigma^2}{\kappa}}$$

*where $D^1_{\text{MSE}}$ is a constant that depends on $(X_i, Y_i, \varepsilon_i)_{i \in I_p}$, $\theta^\star_\sigma$ is the noisy estimate, and $D^2_{\text{MSE}} > 0$.*

*Proof.* First, denote the loss functions generically as $\ell_\theta$. The proof is based on a classical argument, and we show it for WBM. Specifically, strong convexity implies for $\theta, \theta^\star \in \mathbb{R}^d$

$$\ell_\theta \geq \ell_{\theta^\star} + \frac{\kappa}{2} \|\theta - \theta^\star\|^2$$

at the minimizers

$$\ell_{\theta_\sigma^\star}(\sigma) \le \ell_{\theta_\sigma^\star}(0) + \delta \quad \text{and} \quad \ell_{\theta_{\text{WBM}}^\star}(0) \le \ell_{\theta_{\text{WBM}}^\star}(\sigma) + \delta \qquad \text{for } \sigma > 0 \,.$$

By combining these inequalities and the fact that

$$|\ell_{\theta_\sigma^\star}(\sigma) - \ell_{\theta_{\text{WBM}}^\star}(0)| \le D_{\text{WBM}}^1 \sigma + D_{\text{WBM}}^2 \sigma^2 \,,$$

one gets

$$\kappa \left\| \theta_\sigma^\star - \theta_{\text{MSE}}^\star \right\|^2 \le 2(D_{\text{WBM}}^1 \sigma + D_{\text{WBM}}^2 \sigma^2)$$

which concludes the argument.

$\square$

## F  ADDITIONAL NUMERICAL RESULTS

We illustrate the evolution of the WBM loss for different Gaussian noise levels for the polynomial function

$$x \mapsto (x - 2)(x - 4)(x - 5)$$

as opposed to the MSE in figure 10. We note that the WBM loss changes more slowly with noise levels, suggesting it recovers regression functions that are less impacted by noise. Additionally, we

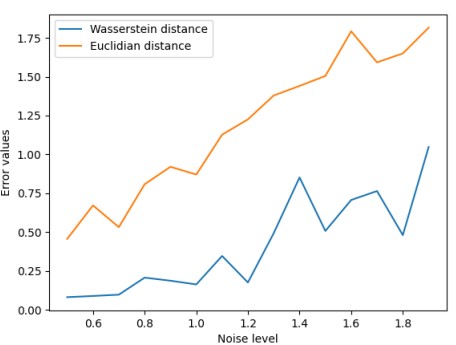

Figure 10: Loss evolution between a function and its noisy version

report error bars over the test batches for the results presented in section 6 as well as complementary numerical results for learning of the Navier-Stokes equation, Wave equation and Electricity Times Series. Note that for training on noisy the error bounds are much smaller due to the fact that the model training smoothens the noise, whereas for testing on noisy data the difference can be significant among batches.

**Batch Size Scaling**
We report in figure 20 the relative test error for various batch sizes, under noise level 1. We note that WBM is not very sensitive to batch size values especially in the smaller range (up to 80).

**Wall-Clock Time** We report in figure 21 the wall-clock time for WBM and the state-of-the-art $\chi^2$-DORO method across batch sizes, for the Navier-Stokes problem and 100 epoch of training. We note that WBM does not require higher time, since the cost is dominated by the dimension as discussed in the complexity remark of section 4. We also report wall-clock time across dimensions in figure 24 demonstrating the dominance of the dimension for time complexity. Furthermore, we report in table **??** below the run time for each class of methods without hyper-parameter tuning. Since WBM does not require any additional hyper-parameter tuning beyond the standard deep learning hyper-parameters (such as batch size), it results in reduction of computational cost of at least an order of magnitude compared to the state-of-the-art which requires at least one additional hyper-parameter.

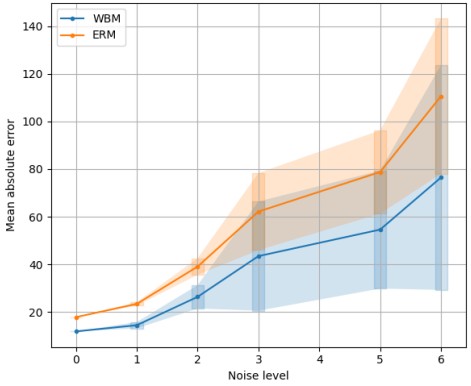

Figure 11: Test error evolution for Navier-Stokes operator learning - 30% corrupted test data with Cauchy noise

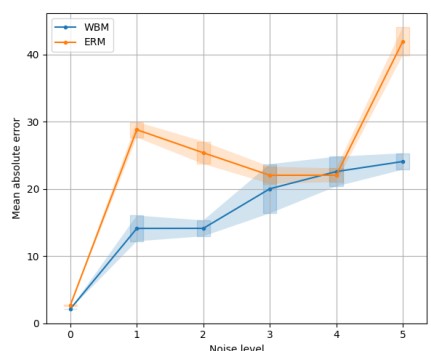

Figure 12: Test error evolution for wave equation operator learning - 30% corrupted training data with Gaussian noise

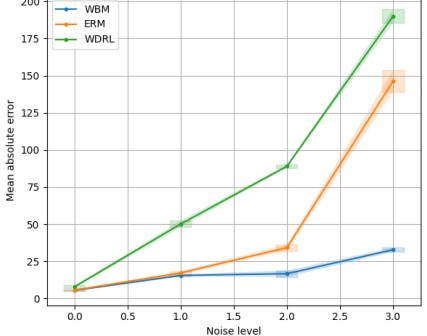

Figure 13: Test error evolution for Navier-Stokes operator learning - 30% corrupted training data with Cauchy noise, $4\sigma$

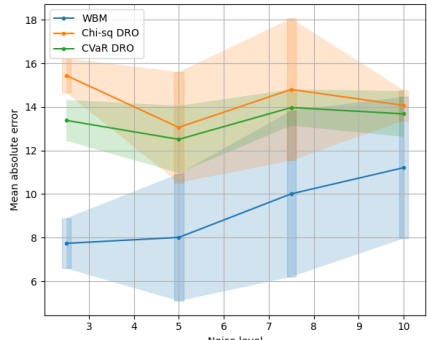

Figure 14: Test error evolution for Navier-Stokes operator learning - 30% corrupted training data with Gaussian noise

**Multiplicative noise**

We explore the effectiveness of WBM in case where the noise is multiplicative, for the Navier-Stokes problem. We report the relative MAE in figure 23. We observe that WBM maintains a relatively stable error across noise levels.

**Correlated noise**

Similarly, we report the relative error of WBM for the Navier-Stokes problem, with strongly correlated noise. The noise is generated by sampling a Gaussian with covariance matrix of the form $\Sigma + \alpha \mathbf{1}\mathbf{1}^{\top}$, where $\alpha$ control the correlation level and $\Sigma$ follows a Wishart distribution with diagonal scaling matrix. We note that WBM is less effective for correlated noise, which is an expected effect given that WBM is based on distributional matching.

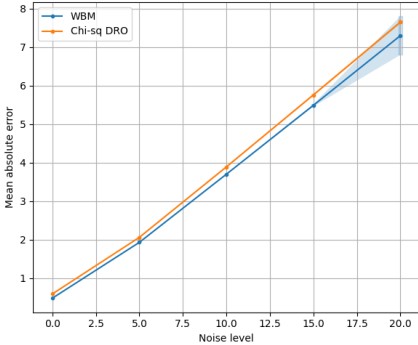

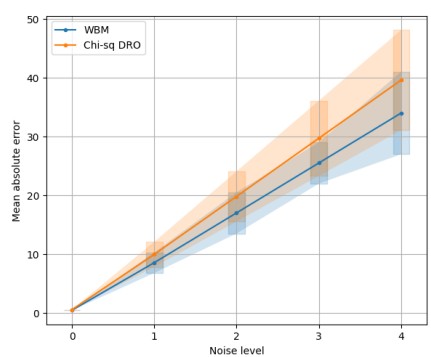

Figure 15: Test error evolution for electric time series forecasting - 30% corrupted test data with Cauchy noise

Figure 16: Test error evolution for electric time series forecasting - 30% corrupted test data with Gaussian noise

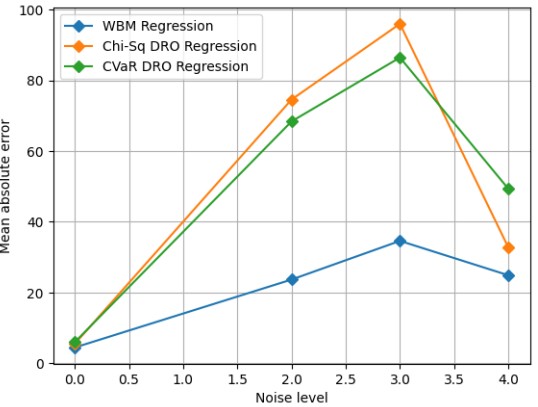

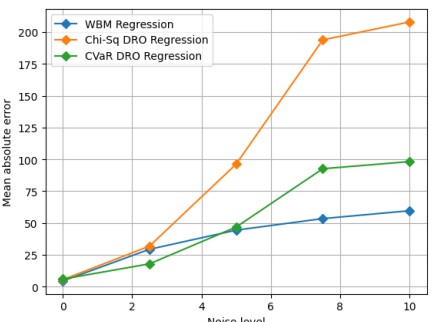

Figure 17: Test error evolution for Navier-Stokes operator learning - 30% corrupted training data with Cauchy noise

Figure 18: Test error evolution for Navier-Stokes operator learning - 30% corrupted test data with Gaussian noise

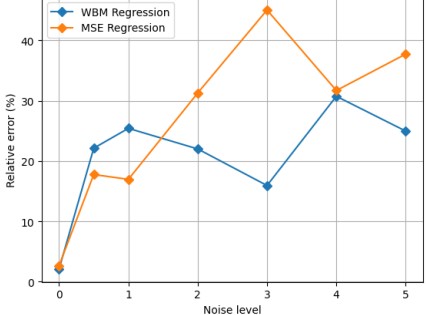

Figure 19: Test error evolution for wave equation operator learning - 30% corrupted training data with Cauchy noise

## G    REPRODUCIBILITY

We provide a version of the code used for the numerical experiments in the following link: code. It is based on modifications of the publicly available code from (Raonic et al., 2023) and (Ilbert et al., 2024). The experiments were performed on NVIDIA Tesla T4 and Tesla P100 16GB Cores.

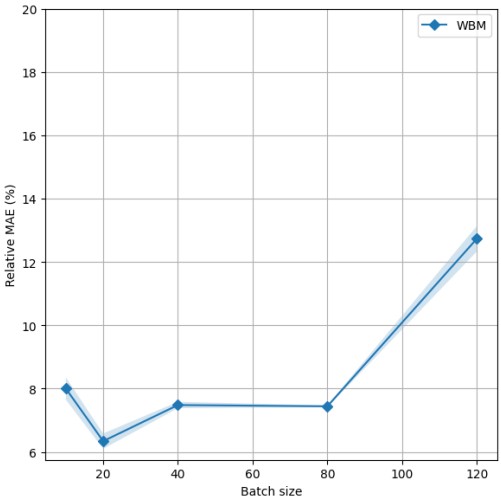

Figure 20: Sensitivity of WBM to batch size

| Method | WBM | $\chi^2$-DORO | WDRL |
|---|---|---|---|
| Time (s) | **2160.92** | 2211.51 | 3476.27 |

Table 1: Execution times of different methods.

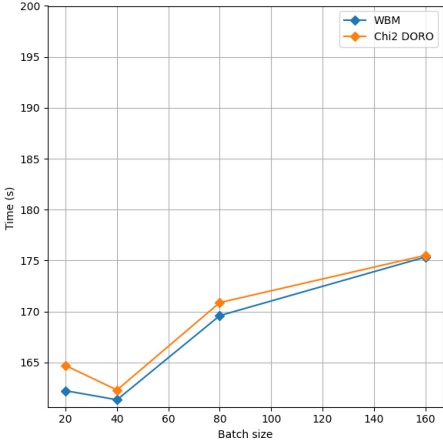

Figure 21: Wall-Clock time across batch size

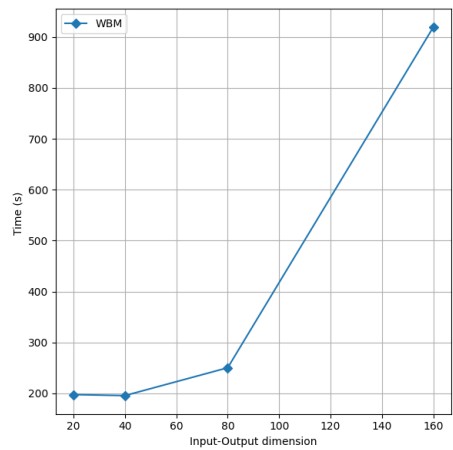

Figure 22: Wall-Clock time across dimensions

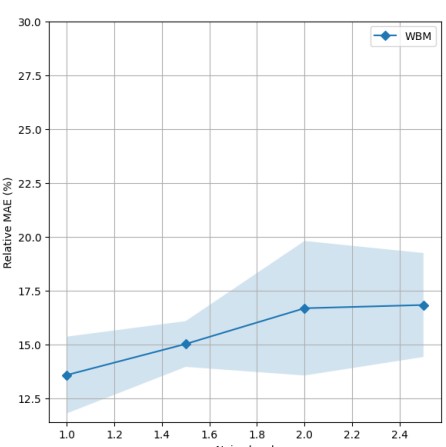

Figure 23: Test error across multiplicative noise levels

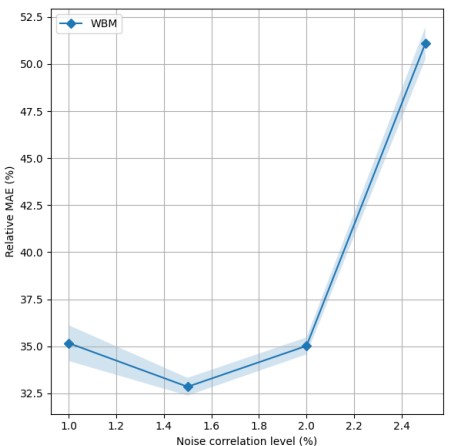

Figure 24: Test error across noise corelation levels

